# Exploring Spatial Distributions of Land Use and Land Cover Change in Fire-Affected Areas of Miombo Woodlands of the Beira Corridor, Central Mozambique

**Victorino Américo Buramuge** [1,*] , **Natasha Sofia Ribeiro** [1] , **Lennart Olsson** [2] and **Romana Rombe Bandeira** [1]

1   Department of Forest Engineering, Faculty of Agronomy and Forest Engineering, Eduardo Mondlane University, Maputo 257, Mozambique
2   Centre for Sustainability Studies (LUCSUS), Lund University, P.O. Box 170, 22100 Lund, Sweden
*   Correspondence: vburamuge@gmail.com; Tel.: +258-84-911-8588

**Abstract:** Miombo woodlands (MW) are increasingly experiencing widespread land use and land cover change (LULCC). This study explores the influence of fire, agriculture, and slope variability on LULCC in the miombo of the Beira Corridor. Land use and land cover data were derived from three Landsat images for 2001, 2008, and 2018. Slope attributes were derived from the Shuttle Radar Topography Mission (SRTM). Monthly burned data of Moderate-Resolution Imaging Spectroradiometer (MODIS) were used to map fire frequency. The derived data were then used to investigate the relationship between LULCC and fire, agriculture, and slope, based on geographically weighted regression (GWR). In addition, the relationship between LULCC and slope was assessed. Our findings indicate that fire frequency, agriculture, and slope were significantly spatially non-stationary. We found that LULCC was negatively correlated with agriculture in open miombo, but positively correlated in dense miombo. A positive relationship between LULCC and fire was found for dense and open miombo. Changes in agriculture, dense miombo, and open miombo increased towards high slopes. The study improves the understanding of the spatial effect of LULCC drivers. The development and implementation of effective fire management actions is required to promote sustainable forest management and preservation of critical ecosystem services.

**Keywords:** fire frequency; agriculture; slope; GWR

## 1. Introduction

Land use and land cover change (LULCC) has become a major process in Southern Africa, with an unprecedented conversion of natural forests over the last decades [1–3]. Miombo woodlands are the most important tropical dry forest type in Sub-Saharan Africa, covering about 2 million km$^2$ [4]. Human activities, fire, elephant populations, climate, and soils are the main factors that determine the dynamics of miombo woodlands [5]. These types of forests are considered resilient as a result of their rapid regeneration following disturbances [6]. Their high regeneration capacity is mainly due to vegetative reproduction through root or stump sprouting conferring high resistance to disturbance effects of fire and human activities [7,8]. Areas of miombo woodlands that are disturbed by charcoal production and shifting agriculture can quickly regenerate, maintaining higher tree density than that of mature miombo woodland [7]. The rapid recovery of miombo woodlands following disturbance is critical for biodiversity conservation and thus, climate regulation.

Despite their high resilience, miombo woodlands are under increased pressure. Rapid population growth in countries containing miombo woodlands (estimated at 2.76%), increases human access to previously remote forest areas, promoting the rapid depletion of essential resources [4] and modification of the landscape [9,10]. Vulnerable populations rely on resources provided by the forest as their main source of subsistence. In Sub-Saharan Africa, poverty is estimated to have declined slightly from 53% to 51% between 1981 and

2005 [11]. However, this slight reduction does not imply a reduction in dependence on natural resources, given the increasing needs of the growing population. Growing resource demand has been identified as a primary driver of forest loss and degradation [12].

Shifting cultivation and fires are among the key determinants of land cover change in the miombo woodlands [13]. Shifting cultivation usually involves the complete removal of vegetation followed by cultivation. When the soil fertility is depleted, the area is abandoned and the natural regeneration of the woodlands initiates [7,14]. Since this agricultural practice uses fire as the main tool to clear the target forest areas, it increases fire frequency [15,16]. Increased fire frequency changes the fire regime [10,17]. Fire may catalyze a series of large-scale ecological changes, changing landscape patterns [18] or promoting the transition from one vegetation type to another [19]. The impact of fire may also involve the loss and degradation of terrestrial ecosystems and may affect the composition, distribution, and structure of vegetation [20].

Depending on the type of ecosystem and its adaptation, the correlation between fires and land cover change can vary significantly, particularly in tropical regions, where fire frequency tends to be very high [10,21,22]. In miombo woodlands, the impact of fire depends on the fire intensity and time of plant phenology [5]. Fires occurring at the end of the dry season tend to be more intense and destructive than those occurring early in the dry season, when much of the vegetation is green and wet [5,23]. Frequent fires in the late dry season can transform dense forests into open stands, promoting grass biomass to increase by up to 50%, increasing the risk of fire occurrence [24]. Post-fire effects may persist for several decades. The recovery of fire-affected areas also depends on the type and condition of affected vegetation, the fire severity, and the land use history [25]. Long intervals between fires generally tend to favor woody plants, particularly under high rainfall on soils amenable to tree growth [5].

Limited human accessibility in high slopes, maintain high forest cover [26] and may prevent landscape fragmentation against degradation agents [27]. Apart from influencing the accessibility, slope is an important factor determining fire behavior. In the presence of winds, fires can propagate easier and faster upslope than on flat land [28].

It has been shown that LULCC drivers act differently spatially and temporally [3], suggesting that the complex spatial diversity in the landscape determines different types of responses [29,30]. Changes occurring in the landscape may eventually trigger a mosaic of patches responding differently to the factors involved in the conversion of forest cover. Despite the known variety of effects of environmental and socio-economic agents causing forest loss, their associated impacts on the landscape variability are not yet well understood.

In Mozambique, the decline of miombo woodland has shown an increasing trend, estimated to be 0.24% in 1994 [31], 0.54% in 2007 [32], and 0.79% in 2016 [13]. According to the most recent national forest inventory, about 267,000 ha of forest is converted annually for shifting agriculture, settlement and infrastructure expansion, logging operations, and firewood and charcoal production [13].

The present study examines the spatial distribution of LULCC in fire-affected areas over the period of 2001–2018 within the Beira Corridor, Central Mozambique. The Beira Corridor is one of Mozambique's six development corridors, linking Mozambique to Zimbabwe and other hinterland countries such as Malawi and Zambia, and as such it has one of the highest deforestation rates in the country (1.8% annually against 0.79% at the national level) [13]. In the Beira Corridor, land use and cover are very dynamic, with shifting cultivation and charcoal production representing the main drivers of deforestation. Fire is used as an important management tool for several human activities.

In this study, we employed local-level statistical approaches to investigate how fire frequency, agriculture, and slope affect the land cover change of miombo woodlands. To our knowledge, this is the first study addressing the spatial relationship between these factors and land cover change in miombo woodland of the study area. The study area is located within a region of extreme poverty, with the poverty index estimated to be around 50% [33]. These populations rely on forests as the main sources of ecosystem goods

and services to support their livelihoods [34]. The high poverty index, especially in rural areas, has been considered one of the main indirect causes of land degradation [35] and deforestation [3].

The study improves the understanding of the spatial effect of LULCC drivers in order to inform the development and implementation of effective fire management actions, that are capable of promoting sustainable forest management and the preservation of critical ecosystem services.

## 2. Materials and Methods

### 2.1. Study Area Description

The study was conducted in the miombo woodlands of Manica district, central Mozambique. This district is part of the Beira Development Corridor, one of Mozambique's six development corridors, linking Mozambique to Zimbabwe and other hinterland countries such as, Malawi and Zambia. The corridor integrates two provinces of central Mozambique, namely, Sofala (Nhamatanda and Dondo districts) and Manica (Gondola and Manica districts). Manica district covers an area of 4594 km$^2$ and borders Barué district in the north, Sussundenga district in the south, and Gondola district in the east. The west is bordered by the Republic of Zimbabwe, characterized by mountains, extending from the south to north of the district [36] (Figure 1).

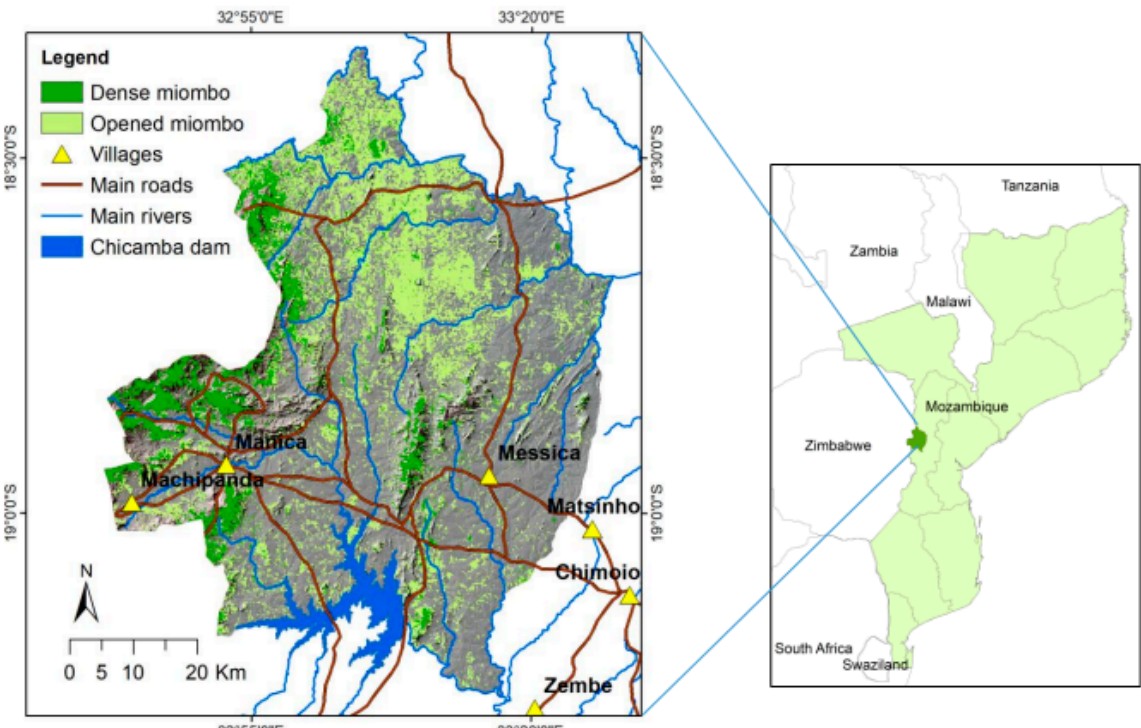

**Figure 1.** Location of the study area, in the center of the country, showing the distribution of dense and open miombo woodlands. Land use and land cover data source: Ministry of Land, Environment and Rural Development [13].

Manica district had about 220,000 inhabitants in 2017, with a density of 76 inhabitants per km$^2$ [37]. Regarding hydrography, the district is drained by the Revué River and its tributaries, which drain into the Búzi River. The topography is gently undulating, with mountains sometimes reaching altitudes of 1500–2000 m. The soils are reddish brown and deep, have low fertility, and are susceptible to erosion. According to the Köppen classification, the climate is tropical–humid, with mean annual rainfall ranging from 1000 to 1020 mm. The rainy season lasts from December to April. The mean annual temperature is 21.2 °C, with a maximum temperature of 30.9 °C and a minimum temperature of 14.0 °C [36].

Manica province is rich in floristic diversity, with mosaics of dense forests, open forests, thickets, and shrubs [32]. Forest formations occupy about 27% of the province's territory. Manica district is covered by semi-deciduous forests, including miombo woodlands [13] and other forest formations under the strong influence of shifting agriculture [32]. The miombo woodlands are known for their high species diversity [4], and the most dominant subfamily is Detarioideae, consisting of three genera, namely, *Brachystegia*, *Julbernadia*, and *Isoberlinia* [38]. The miombo woodlands are mainly found in the district's north and central regions and are an important source of income for the rural population [34].

### 2.2. Data Acquisition

2.2.1. Fire Frequency Mapping

For fire frequency data, we used the Moderate Resolution Imaging Spectroradiometer (MODIS) dataset, specifically, the monthly burned area product (Level-3 gridded 500 m MCD64A1), gathered from https://earthexplorer.usgs.gov/ (accessed on 11 June 2019). All data are from 2001 to 2018. The MODIS MCD64A1 product contains a monthly gridded product of burned areas at a 500 m resolution that combines the burned areas from Terra and Aqua MODIS. The MCD64A1 is an improved version of the MCD45A1 collection 5.1, adapted to different conditions in various ecosystems [39].

We converted the monthly burnt area data from HDF to GEOTIF format in ArcGIS 10.1 (ESRI, Redlands, CA, USA). We then successively summarized the monthly burned area product to produce a burned and non-burned layer. Next, we aggregated the derived monthly layer to generate an annual dataset. Finally, the resulting yearly burned area layers were aggregated into a final map of fire frequency for 18 years.

To validate the fire frequency map, an accuracy assessment exercise was performed using the error matrix method [40]. We also relied on field based reference data for accuracy assessment to validate the classification results [41]. The evaluation was carried out in 107 plots over two seasons: dry (October) and wet (April). For each sampled area, visual observations were conducted of the signs of fire occurrence, such as charred litter, partially or wholly consumed stems and trunks, and indications of a charred soil layer [42]. We computed the metrics of overall accuracy, user accuracy, and the Kappa statistic.

2.2.2. Land Use and Land Cover Change Mapping and Slope Data

In this study, we used a set of two Landsat 5 (TM) and one Landsat 8 (OLI/TIRS) multi-temporal images, Path/Row 168/73, corresponding to 10 August 2001, 29 August 2008, and 28 October 2018, to analyze the LULCC for the miombo woodlands of the study area. All images were acquired from the United States Geological Survey (USGS; http://earthexplorer.usgs.gov; accessed on 6 June 2021), during the dry season, so as to minimize seasonality and cloud effects. After delimiting the thematic classes, the multivariate maximum likelihood classifier [43] was applied for the main land use and land cover (LULC) types present in the area: (i) dense miombo, (ii) open miombo, (iii) agriculture, (iv) water bodies, and (v) other classes. The slope data were derived from Shuttle Radar Topography Mission (SRTM) at a 30 m spatial resolution, obtained from https://earthdata.nasa.gov (accessed on 10 February 2021).

The LULC thematic map accuracy was assessed using an error matrix for the producer's accuracy, user's accuracy, overall accuracy, and the Kappa statistic coefficient (Kappa) [40]. For this analysis, a stratified random sampling of pixels was drawn from the map. The number of pixels per class was based on the proportion of the LULC class pixel frequency against the total number of mapped pixels. A total of 396 random samples were generated in the classified image of the study area. Each random point was visually interpreted using historical high-resolution Google Earth images and the national inventory [13] to generate error matrices for classified maps. The field observation was also conducted at two periods—dry (October) and wet (April)—to assess the LULC types that existed across the study area. We assessed the feature and patterns of land use and land cover of the study area, such as type of vegetation, slope aspect, agriculture areas, fallow land (regrowth

areas), land occupation, and other biophysical characteristics. A total of 60 reference points were collected during field work for LULC classification accuracy assessment.

### 2.3. Data Analysis

This study presents an exploratory analysis of the spatial pattern of LULCC in fire-affected areas of miombo woodlands. Considering the specific context of the study area, and the potential to cause changes in the miombo area, three predictor variables were assumed (fire frequency, agriculture, and slope for the two types of miombo woodlands (dense and open miombo).

We conducted separate analyses for dense and open miombo woodlands, given that factors associated with human activities act differently in the two types of miombo. For instance, the dense miombo occurs mainly in high-elevation areas (up to 1600 m), where human accessibility is limited [27,44]. The elevation is also considered an important factor in fire occurrence and land cover dynamics [45,46], and a key limiting factor of human accessibility [47].

The derived land use and land cover map in raster format, was converted into a vector map and intersected with $1 \times 1$ km grid cells, derived from the MODIS burned areas vector of the study area. The predictor variables (fire frequency, agriculture, and slope) were systematically extracted in the $1 \times 1$ km grid cells. The determination of a suitable cell size for data analysis depends on a set of factors including computer processing power, size of delineations, spatial autocorrelation of the variables, and complexity of the terrain [48]. Therefore, the grid size may also be selected by comparing the results of the various sizes of the grids tested [49]. Three grid sizes were tested: 500, 750, and 1000 m. The 1000 m grid generated more accurate results for the parameters under consideration (Table S3).

Based on the data generated in the grid cells, the predictor's variables were tested using conventional ordinary least squares (OLS) to evaluate the global relation between explanatory variables and LULCC as the dependent variable. In a global regression (Equation (1)), the regression model parameters are considered stationary in the analysis space. This model assumes that each point in the study area is absolutely representative and the quantified relationship is constant [50]. The global regression model [51] can be represented as:

$$yi = \alpha + \beta x_{i1} + \ldots + \tau x_{in} + \varepsilon_i \tag{1}$$

where $y$ denotes the dependent variable; $x$ is the independent variable; $\alpha$, $\beta$, $\ldots$ $\tau$ represent the parameters to be estimated; $\varepsilon_i$ represents an error term; and $i$ represents a point in space.

The multicollinearity in the OLS model was assessed through the variance inflation factor (VIF). The overall model significance was assessed using the Joint Wald Statistic. We used the AICc as the goodness-of-fit criteria for model comparisons. The Akaike Information Criterion (AIC) is a metric that assesses the statistical quality of a model. According to this criterion, lower AIC values represent better quality and simplicity of the model [52].

The spatial independence of residuals of the model was evaluated using the spatial autocorrelation coefficient of Moran's I. Moran's I ranges from 0 to 1 for positive autocorrelation and between 0 and −1 for negative autocorrelation, while 0 indicates randomness. Statistically significant spatial autocorrelation of residuals may result in model misspecification [53]. A suitable model may have a random distribution of the residuals [54]. We assessed the model stationarity by applying the Koenker (BP) statistic (Koenker's Studentized Bruesch–Pagan statistic) to determine whether the explanatory variables in the model exhibited spatial non-stationarity. When spatial non-stationarity was observed, we applied the geographically weighted regression (GWR) model. Maps generated by GWR models were used as valuable tools to interpret the phenomena [55].

GWR is non-stationary technique that models spatially varying relationships. Although the GWR model is not adjusted for strictly local observations, it can be adjusted at each observation location in the dataset [56], since the parameters of the independent variables to be estimated are a function of the spatial location [57]. In this type of model,

the parameter estimation is conducted using an approach in which the contribution of a sample to the analysis, is weighted based on its spatial proximity to the specific location under consideration [58]. The GWR method uses the coordinate of the observation point to determine the weighting value given to each parameter. The closer the distance between the parameter and the observation point, the greater the weight value assigned to that parameter [59]:

The GWR model can be represented as follows [52]:

$$y_i = a_0 + \sum_k a_k X_{ik} + \varepsilon_i \tag{2}$$

where $y_i$ is the dependent variable at location $i$, $X_{ik}$ is the independent variable at location $i$, $a_o$ is the intercept parameter at location $i$, $a_k$ is the local regression coefficient for the $ik$ independent variable at location $i$, and $\varepsilon i$ is the random error term at location $i$.

The GWR model in Equation (2) can be rewritten as:

$$y_i = a_0(u_i, \ v_i) + \sum_k a_k(u_i, v_i) X_{ik} + \varepsilon_i \tag{3}$$

where $(u_i, v_i)$ represents the coordinates of the $i$th point in space, and $a_k(u, v)$ is a realization of the continuous function $a_k(u, v)$ at point $i$.

Relationship between LULCC and Slope

In addition, we examined how slope gradients affect the LULCC distribution for agriculture, specifically, dense and open miombo woodlands. The slope in the study area ranges between 0 and 55°. To explore this relationship, the derived slope maps were reclassified into four categories according to the slope classification of the study area, as follows [60]: flat to gentle slope (0–7°), moderate slope (8–13°), moderately steep slope (14–20°), and very steep slope (21–55°). The classified land use and land cover maps of each period (2001–2008, 2008–2018, and 2001–2018), were overlaid on the slope to obtain the relationship between land cover type distribution and changes in each category of slope. The changes in slope classes according to the land cover types were tested for normal distribution with the Shapiro–Wilk test. To compare the means of the variables between different slope categories, we used the Kruskal–Wallis test for non-normally distributed data, followed by Dunn's test at the 5% significance level.

## 3. Results

### 3.1. Accuracy Assessment

Accuracy assessment was performed to determine the agreement of produced classification with what actually exists on the ground [40]. The overall accuracies for land use and land cover classification for the different time periods were 88% for 2001 and 2008; and 86% for 2018, which express a very good agreement between the mapping exercise and the actual situation on the ground. The Kappa index showed excellent agreement. The Kappa index evaluates to what extent a classification is different from a random distribution, and we found 0.82 for 2001, 0.81 for 2008 and 0.86 for 2018, implying a very good result. The validation of the burned areas' MODIS product, based on confusion matrices, resulted in an overall accuracy of 94.7% and Kappa statistic of 0.84. The producer's accuracy in detecting burned-area pixels was 81.3%, where 18.7% of all burned pixels were wrongly classified as non-burned.

### 3.2. Burned Area and Land Cover Change

Our results show that fire was more prevalent in the north of the district (Figure S2a), where the ignitions increased between 2001 and 2008, then decreased from 2008 to 2018 (Figure 2). The peak of fire occurrence occurred in 2008. Overall, the results show a significant conversion of forest cover to agriculture, one of the main subsistence activities in the study area (Figure S1). Between 2001 and 2018, agricultural area increased by 78.9%, while

open miombo decreased by 52.1%, followed by dense miombo (18.2%). Dense miombo showed a decreasing trend for 2001–2008 (10.5%) and 2008–2018 (31.1%). Nevertheless, open miombo cover increased from 2001 to 2008 (19.4%) and decreased between 2008 and 2018 (52.2%).

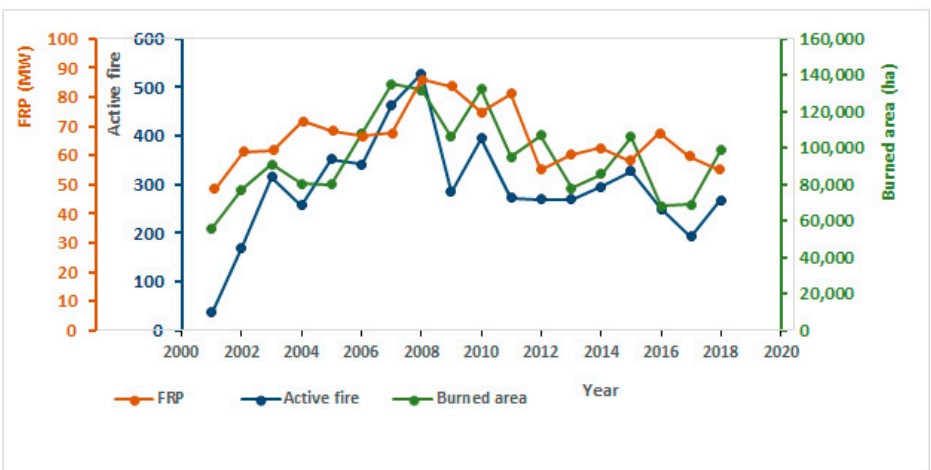

**Figure 2.** Spatio-temporal distribution patterns of active fires (**blue line**) showing the intensity (**red line**) and burned area (**green line**) of miombo woodlands in the study area from 2001 to 2018.

*3.3. OLS Regression*

Table 1 shows the results of the OLS regression model for dense and open miombo. We further examined the residuals of the OLS model for both types of miombo woodlands and found that the residuals had a strong positive spatial autocorrelation, presenting a clustered pattern for dense miombo (Moran's I = 0.249, $p$ < 0.001) and for open miombo (Moran's I = 0.295, $p$ < 0.001). The presence of autocorrelation in a model's residuals indicates that the model is mis-specified [54]. Moreover, the Koenker (BP) statistics for both models indicated that the residuals were not distributed identically, indicating non-stationary variance for dense miombo (Koenker: 175.6, $p$ < 0.00) and open miombo (Koenker: 202.5, $p$ < 0.00) models.

**Table 1.** OLS results for the two types of miombo woodlands.

| Miombo Type | Variable | Coefficient | SE | $p$-Value | VIF |
|---|---|---|---|---|---|
| Dense miombo | Constant | 9.601 | 0.584 | 0.000 | |
| | Agriculture | −0.246 | 0.009 | 0.000 | 1.061 |
| | Fire frequency | 0.004 | 0.014 | 0.000 | 1.059 |
| | Slope | −0.713 | 0.033 | 0.850 | 1.020 |
| **Model Fitting Information** | | | | | |
| R$^2$ | | | | 0.35 | |
| AICc | | | | 14,964.35 | |
| open miombo | Constant | −5.667 | 0.358 | 0.000 | |
| | Agriculture | −0.874 | 0.006 | 0.000 | 1.005 |
| | Fire frequency | −0.003 | 0.010 | 0.785 | 1.012 |
| | Slope | 0.642 | 0.027 | 0.000 | 1.008 |
| **Model fitting information** | | | | | |
| R$^2$ | | | | 0.84 | |
| AICc | | | | 28,626.59 | |

The VIF values were acceptable and varied from 1.003 to 1.006, indicating that OLS estimations for both types of miombo woodlands were not biased from multicollinearity [61] (Table S4). The presence of multicollinearity indicates that there are high correlations between predictor variables in the regression model and constitutes a limitation, because it affects the statistical significance of the predictor variable [62], increasing the standard errors of coefficients in the model [61]. The examination of the overall models indicated a statistical significance model for dense miombo (Joint Wald: 563.6, $p < 0.001$) and for open miombo (Joint Wald: 14,248.9, $p < 0.001$).

### 3.4. GWR Model and Spatial Variations

As a result of the existence of the high spatial autocorrelation of residuals, violating the assumptions of OLS estimation and an indication of non-stationarity, we used a GWR model to fit the data for the two types of miombo woodlands. Figure 3 shows the spatial distribution of the standardized residuals for dense and open miombo for the study area. For both types of miombo, the standardized residuals appear to be randomly distributed. This model was more efficient than OLS, by improving the explanatory power (R2), AICc and reducing the spatial autocorrelation for the residuals (Table 2).

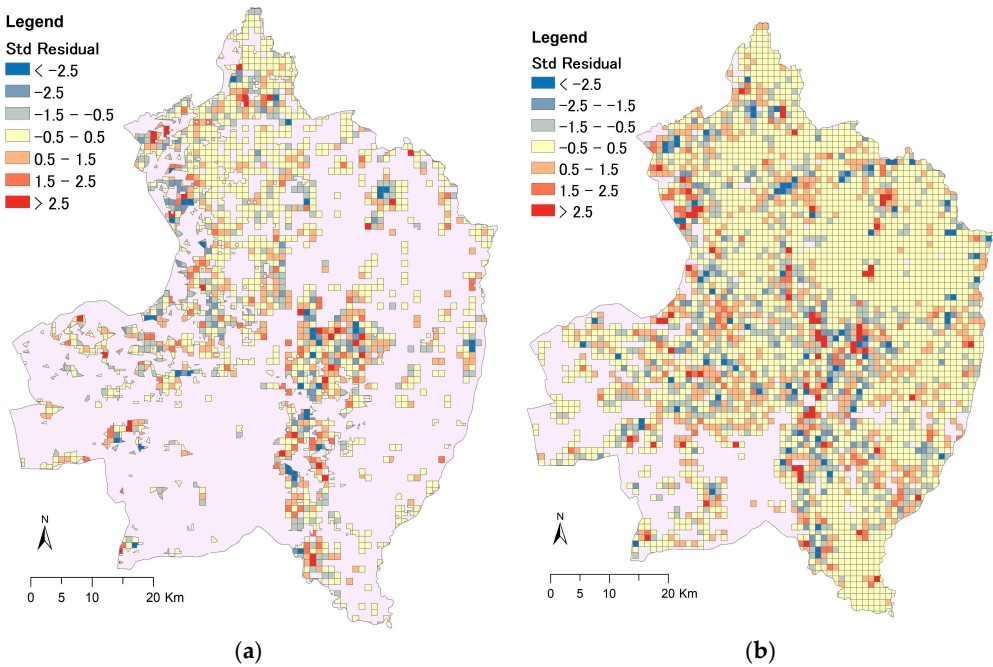

(**a**)          (**b**)

**Figure 3.** GWR local spatial distribution of standardized residuals for dense (**a**) and open (**b**) miombo woodland.

**Table 2.** Residuals of the spatial autocorrelation and GWR fit statistics from the two types of miombo woodlands.

| Miombo Type | Residuals | | | | |
|---|---|---|---|---|---|
| | Moran's I | z-Score | *p*-Value | AICc | Adjusted R$^2$ |
| Dense miombo | 0.001 | 0.259 | 0.796 | 3739.38 | 0.67 |
| open miombo | 0.008 | 1.562 | 0.118 | 4514.36 | 0.93 |

The GWR model takes into account the non-stationarity of the relationship between spatial variables, allowing the estimation of parameters for each specific point [51]. In this method, we estimated the parameters, taking into account not only the predictive variables, but also the Gaussian kernel function with adaptive kernel bandwidth.

The goodness of model fit was evaluated as a function of optimal bandwidth, with the overall model explanatory power ($R^2$) and the Akaike Information Criterion (AIC). Visual observation of the GWR residuals for dense miombo (Figure 3a) and open miombo (Figure 3b), suggests an even distribution. The results established the effectiveness of the GWR as a modeling tool for the LULCC dynamics of the miombo woodlands in the study area. The local multicollinearity ranged from 3.353 to 11.401 for dense miombo (Figure 4a), while for open miombo (Figure 4b), the values varied from 3.072 to 23.543, indicating the absence of multicollinearity between the predictor variables for both models. A summary of the GWR model results is presented in Table 2.

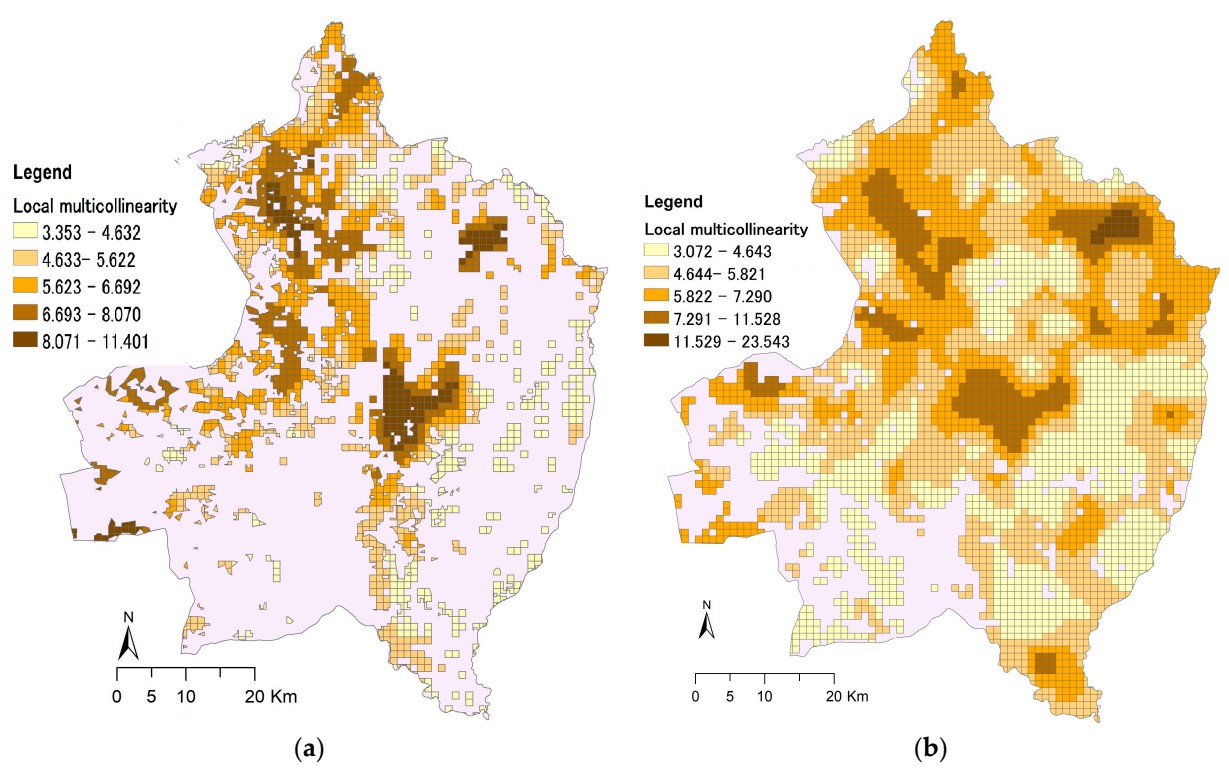

**Figure 4.** Map of the GWR local multicollinearity among predictor variables for dense (**a**) and open (**b**) miombo woodlands.

The relationships between LULCC and the predictor variables for dense and open miombo woodlands are shown in Figures 5 and 6, respectively. The GWR model performance is spatially heterogeneous for the study area in both dense and open miombo. The GWR model manifests different positive and negative relationships across the study area. The global explanatory power observed was very high for open miombo (Adjusted $R^2$ = 0.93), varying locally from 0.02 to 0.99, while for dense miombo, the adjusted $R^2$ was 0.67, varying locally between 0.01 and 0.95.

The dense miombo is best explained in the central and southern part of the study area (Figure 5a). This pattern can also be observed along the main roads connecting the district to other areas, such as Zimbabwe. The correlation between fire frequency and land cover change was strong in the western part of the study area and along the main roads. For agricultural areas, the positive relationship occurred mainly in the western and eastern areas. The same trend was observed along the main roads. Overall, two specific points showed a strong positive correlation between slope and with LULCC (Figure 5d). This relationship can be observed in the mid-northern and southern part of the study area. These areas are crossed by the main roads connecting the study area and other areas.

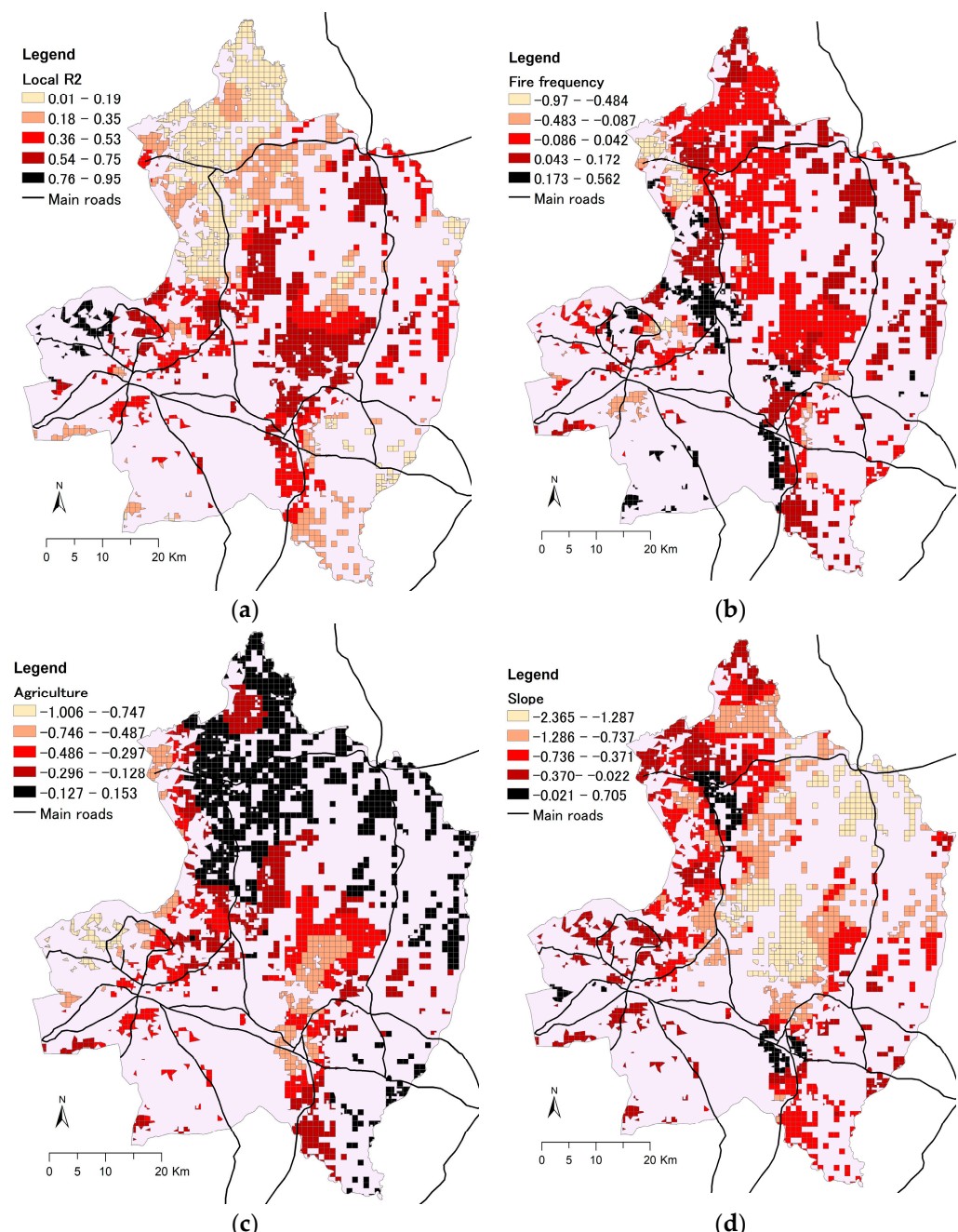

**Figure 5.** Spatial mapping of the locally weighted coefficient of determination ($R^2$): (**a**), fire frequency (**b**), agriculture (**c**), slope (**d**), GWR modeling for dense miombo. The more intense the brown color, the stronger the correlation. The cell sizes are $1000 \times 1000$ m.

For open miombo, the model was more suitable in the central area of the district (Figure 6a). It was also possible to observe high explanatory power in the southern region. The correlation between fire frequency and land cover change was strong in the eastern and western parts of the study area (Figure 6b). Nonetheless, the impact of roads on this relationship is obvious. However, an inverse relationship was found with agriculture. The observed relationship is stronger in the southern area, particularly along the main corridor connecting Zimbabwe (Figure 6c). This region is densely populated and maintains the main infrastructure of the district.

The slope showed a positive relationship across much of the district, mainly in the mid-northern and southern parts that are characterized by high elevations (Figure 6d).

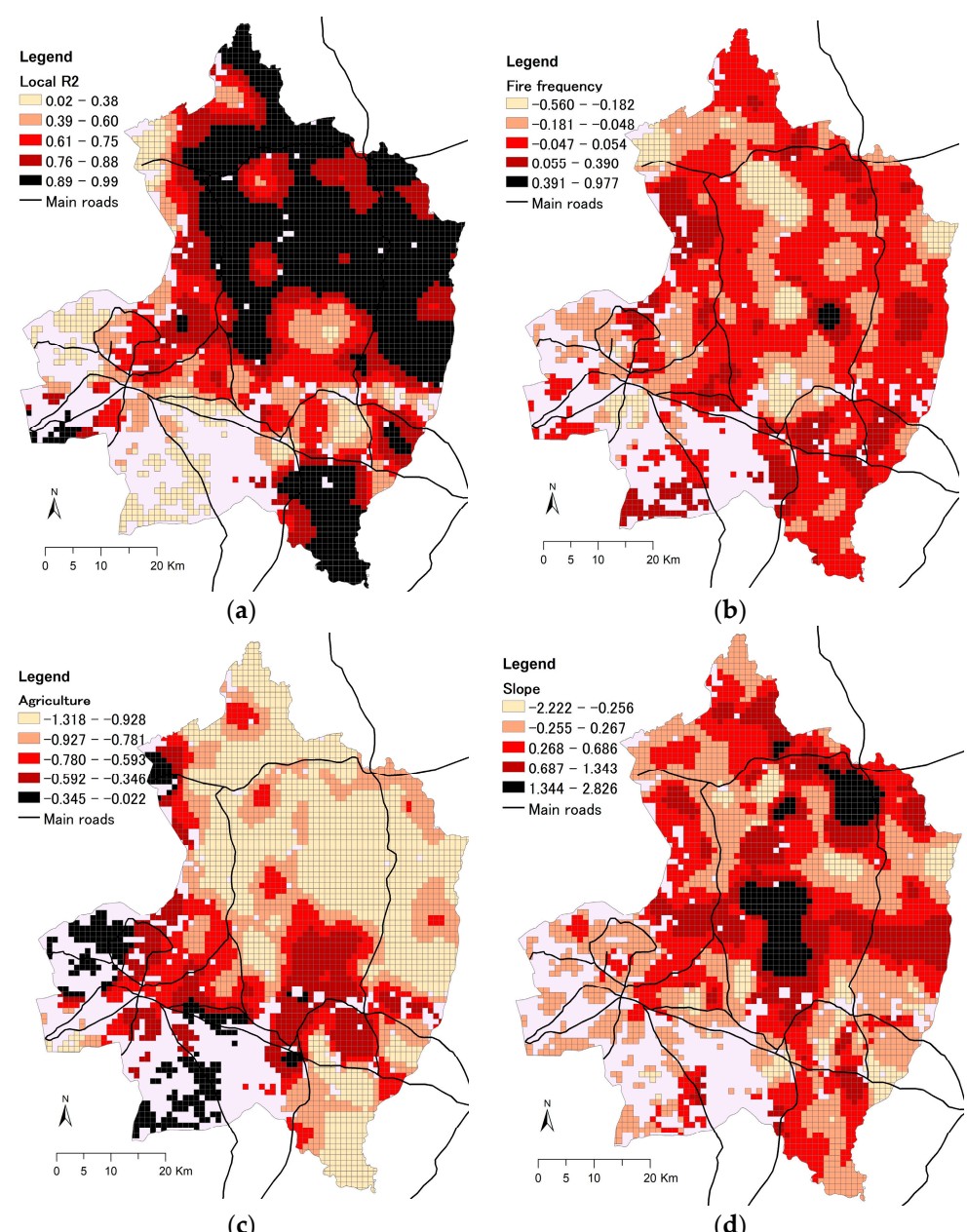

**Figure 6.** Spatial mapping of the locally weighed coefficient of determination ($R^2$): (**a**), fire frequency (**b**), agriculture (**c**), slope (**d**), GWR modeling for open miombo. The more intense the brown color, the stronger the correlation. The cell sizes are $1000 \times 1000$ m.

### 3.5. Land Use and Land Cover Change across Slope

Figure 7 shows the spatial distribution of the different LULC across slope categories. The LULC was strongly affected by the slope gradient. The three land use and land cover types (agriculture, dense and open miombo woodlands) were found in different proportions across the four slope categories. Agricultural expansion was observed on all slope classes (low to high slopes). The low slope category had a significantly higher proportion of change from 2001 to 2008, and the lowest was observed between 2008 and 2018 (Table 3).

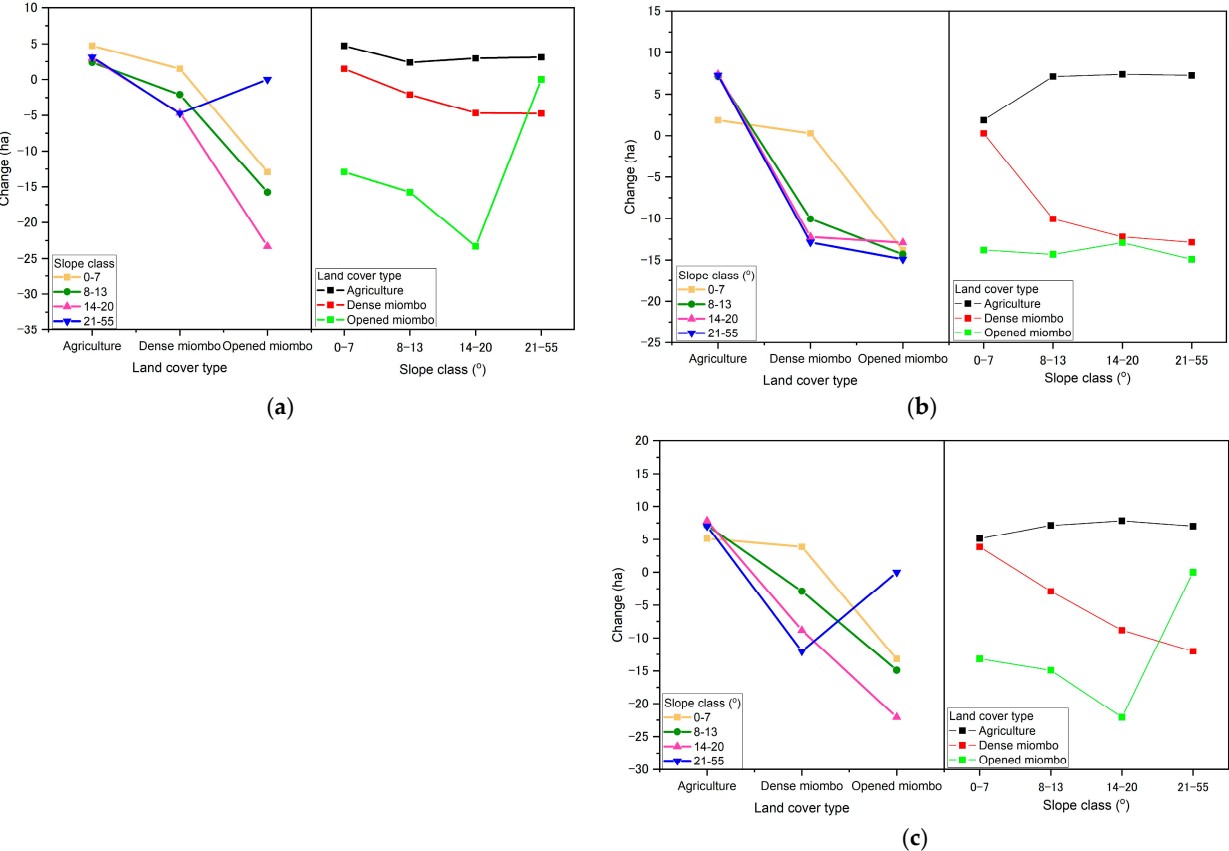

**Figure 7.** Spatial distribution of the different land use land cover changes across slope categories in 2001–2008 (**a**), 2008–2018 (**b**), and 2001–2018 (**c**).

**Table 3.** Land cover change distribution across the slope gradient. Means that do not share a letter are significantly different (Kruskal–Wallis test; $\alpha = 5\%$).

| Land Cover Type | Slope Class | 2001–2008 | 2008–2018 | 2001–2018 |
|---|---|---|---|---|
| Agriculture | Low | 4.7 [a] | 1.9 [b] | 5.1 [b] |
| | Moderate | 2.4 [b] | 7.1 [a] | 7.1 [a] |
| | Steep | 2.9 [b] | 7.4 [a] | 7.8 [a] |
| | High steep | 3.1 [b] | 7.3 [a] | 7.0 [a] |
| Dense miombo | Low | 1.5 [a] | 0.3 [a] | 3.9 [a] |
| | Moderate | −2.1 [b] | −10.1 [b] | −2.9 [b] |
| | Steep | −4.7 [c] | −12.2 [bc] | −8.8 [c] |
| | High steep | −4.8 [c] | −12.9 [c] | −12.0 [d] |
| Opened miombo | Low | −12.9 [a] | −13.8 [a] | −13.2 [a] |
| | Moderate | −15.6 [b] | −14.2 [a] | −14.9 [a] |
| | Steep | −23.3 [ab] | −12.9 [a] | −22.1 [a] |
| | High steep | 0.03 [ab] | −14.9 [a] | −0.2 [a] |
| | *p* | <0.001 | <0.001 | <0.001 |

For the dense miombo, the proportion of change increased with the slope, and a significantly lower change was observed at low slope. In contrast, the change in the open miombo woodlands increased from the 0–7° to the 8–13° slope category in the periods of 2001–2008 (Figure 7a) and 2001–2018 (Figure 7c). For the period of 2001–2008 (Figure 7b), the conversion of a significant part of dense miombo to open miombo was observed, mainly within the 14–20° slope category. The conversion of a high proportion of dense miombo to open miombo was also observed in 2001–2018 from the 14–20° slope class. In contrast, although open miombo woodlands experienced a loss of coverage across the slope

gradient, no significant difference in land cover change was observed among the four slope categories, in 2001–2008 and 2001–2018 (Table 3).

## 4. Discussion

The period of analysis was characterized by a continuous critical loss of forest cover, accompanied by an increase in agricultural expansion. Fire was associated with these dynamics and showed high frequency in the north of the district, which could extensively affect the vegetation. These results reflect a national pattern of fire occurrence, with an increase between 2001 and 2008, followed by a decrease through 2018 [37]. The observed dynamics may be associated with the national historical context of 16 years of civil war. During this period, the lack of security restricted access to rural areas for the extraction of essential resources. Consequently, the population sought safety in nearby countries such as Malawi, Zambia, Tanzania, and Zimbabwe. The peace agreement was signed in October 1992, which marked the end of the civil war and the return of most of the population.

As a result, vital activities including demobilization, demining, refugee repatriation, re-settlement, and physical reconstruction were implemented. Legal and institutional reforms were adopted [63]. One of the key turning points was the establishment of the Ministry for Coordination of Environmental Action (MICOA) in 1994. In sequence, the environmental law was implemented in 1997 and the environmental impact assessment regulation was adopted in 2004, for environmental management and policy formulation. Legal frameworks for the preservation, management, and sustainable use of forest resources were adopted (Forest Law, in 1999). The Ministry for Coordination of Environmental Action produced two environmental education-related documents in 2006: the Communication Strategy for Environmental Education Dissemination (ECODEA) and the Program for Environmental Education, Communication and Dissemination (PECODA). These documents aimed to promote the adoption of new attitudes toward environmental issues, and the environmental communication and education for communities. Meanwhile, in 2008, an intensified environmental education program was implemented to control wildfires [37]. The post-war reforms implemented may have contributed to a decrease in fires between 2008 and 2018. The decrease in fire incidence during this period could also be attributed to forest fragmentation caused by intensive land use [10]. The massive reoccupation of previously abandoned areas increased access to rural areas, leading to the conversion of forests to other uses, such as commercial and shifting agriculture uses and the establishment of infrastructure [64]. Furthermore, extensive human presence in these areas has had a profound effect on the LULCC, with particular emphasis on the use of fire, as one of the main management tools [65].

The pattern of LULCC exhibited a strong spatial variability, which was influenced by different factors. Applying the GWR model allowed us to capture the local variations of the factors involved in the LULCC, with better accuracy than with the OLS model. The low performance of the OLS model may have been influenced by spatial effects, reducing its explanatory power [66]. However, fire mapping using coarse resolution such as the MODIS MCD64A1 product (500 m) is challenging, especially for tropical ecosystems [67]. Agriculture-related burnings that occur in the study area are often small (<5 ha) and therefore difficult to detect [68,69]. The use of high-resolution sensors capable of accurately detecting agricultural burning may overcome these limitations. Nevertheless, many of these sensors do not support systematic data acquisition for time series analysis. Sentinel-2 and Landsat satellites are recognized as potentially critical for future missions to accurately map burned agricultural areas [70].

The observed non-stationarity variables and their influence on the LULCC in the study area revealed that the relationship between variables is not constant. For the two types of miombo woodlands, local positive angular coefficients were observed, indicating that there are strong positive correlations between the predictor variables and the proportion of LULCC. These types of relationship were expected, given that the causes and magnitude of deforestation vary by region and are affected by climatic [71,72] and socio-economic [1,12,73]

contexts. These factors frequently interact, making this relationship extremely complex [3]. Local negative correlations between fire and LULCC were observed across much of the study area, indicating that the effect of fire on dense and open miombo is very limited in some areas. This type of relationship was observed in areas where the vegetation is subject to some type of interference, such as cultivation areas, plantations, settlements and other uses [21]. Areas with intense land use dynamics, habitat loss and landscape fragmentation can occur, reducing the possibility of fire propagation [10].

The spatially simultaneous occurrence of predictor variables involved in the LULCC, indicates that fire occurrence in the region is influenced by humans, due to its strong link to agriculture and other activities, such as charcoal production. These activities are a major source of uncontrolled fire in the miombo [33,65,74]. Due to the relative accessibility of forest areas and the presence of ignition agents, low slopes increase the risk of fire occurrence [75]. Although fire is not considered the main cause of LULCC, in the miombo woodlands, previous studies have shown that its effects can act on forest degradation by influencing tree establishment [76–78] and species composition and diversity [79]. Consequently, changes in forest cover caused by fire may not be immediately reflected in all cases [10].

Slope has been described as one of the most important driving factors of LULCC [26,27,44]. For instance, agriculture frequently occurs in areas with a low slope gradient, due to relative accessibility, whereas steep slopes are more difficult to access and therefore tend to maintain a high proportion of forest cover [26,44,47]. Surprisingly, in the present study, we found that LULCC increased as the slope increased in some highly sloping miombo forests, both open and dense. A high proportion of agricultural areas and fires were also observed. It seems that the scarcity of productive land on the least sloping areas caused increased demand for land towards the high slopes (Table 3), which had previously maintained a high forest cover [80].

Similar patterns of LULCC across high elevations involving agricultural land and urban areas were reported [81]. Other studies [82] noted that agricultural intensification occurred on the higher slopes with dense vegetation, while the expansion was mostly observed on lower slopes. In miombo woodlands, soils with the highest forest cover have been considered to be the most fertile. Shifting agriculture is the main method of cultivation and the main source of income for local communities [14].

Nevertheless, agriculture is unlikely to be the main factor involved in the LULCC found for the open miombo, mainly along the Beira Corridor, across the road connecting Manica district to Zimbabwe. In contrast, the positive correlation with agriculture in dense miombo is clear evidence that agriculture is one of major drivers of deforestation. The corridor's development dynamics have contributed to rapid population growth, resulting in substantial land occupation. Therefore, the main roads and human settlements act as attractors of land occupation [44]. Many of the communities established along the main roads crossing the district rely on agriculture for their livelihood. The immediate consequence is the rapid depletion of available wooded land, especially in areas located along the main roads. To meet the growing demand for land, these areas are often converted to other uses, such as settlements and other types of infrastructure, contributing to settlement sprawl and further increased population density. The process of the occupation of new areas is often accompanied by the establishment of new settlements and, consequently, the practice of subsistence agriculture [33]. With the lack of other job opportunities, this type of land occupation negatively affects the environment and the sustainability of forest resources in particular [83]. The main roads provide access to intact forest areas for commercial logging. The paths open to forest areas are later used for firewood extraction and charcoal production. When the trees are removed, the areas are converted into agricultural land [33]. As fertile land is depleted, the demand for high elevations grows, as evidenced by our findings.

LULCC-related factors may have contributed to forest loss and degradation, mostly in the southern part of the study area, strongly affecting environmental quality and human wellbeing. LULCC is associated with the loss of important ecosystem services, especially

when land occupation occurs in sensitive areas such as high slopes [84]. The loss of ecosystem services reduces the options for the sustainable use of natural resources, particularly for communities highly dependent on them, leading to increased vulnerability to poverty.

Our results show that fires are influenced by human activities. The inappropriate use of fire may result in uncontrolled fires, destroying extensive forest areas. When the fires are repeated, they can deplete seed sources, consequently affecting forest sustainability through regeneration and succession [35]. In this context, fire management actions involving local communities are required to reduce fire hazards, particularly in areas with high forest cover. Since fires in the later dry season are more intense and destructive, prescribed fires have the potential to reduce fuel load, thereby promoting the conservation of miombo woodlands.

## 5. Conclusions

The study area experienced severe forest losses between 2001 and 2018. This period was also characterized by the expansion of agricultural lands. Shifting agriculture is the dominant LU type, and it is one of the main subsistence activities in the study area. The loss of forest cover was accompanied by an increase in fires, which peaked in 2008 and has since declined. Spatial analysis using local-level statistics was useful in demonstrating the LULCC process and associated factors such as agriculture, fire and slope. The relationship was non-stationary for both dense and open miombo areas, due to the high spatial variability and complexity of the factors involved. Spatial heterogeneity and variations in the relationship revealed that LULCC was negatively correlated with agriculture in open miombo, but positively correlated in dense miombo, mainly in the district's northwestern parts. The positive correlation in dense miombo is clear evidence that expanding agricultural areas play a role in miombo woodland loss. The strong negative correlation between LULCC and agriculture in open miombo suggests that forest areas are being converted for other uses, such as settlements and other infrastructure. A positive relationship between LULCC and fire was found in large areas of dense and open miombo. Our findings also revealed the effect of slope on LULCC dynamics. Changes in agriculture, dense miombo and open miombo were found in all four slope categories, increasing towards high slopes. Forest depletion on the lower slope has moved resource demand towards the upper slope, where the proportion of forest is higher and soils are more fertile. Our study provides evidence that fire, agriculture and slope gradient have a locally relevant influence on LULCC.

This study contributes to a better understanding of the spatial effect of LULCC drivers. This is critical for the development and implementation of effective fire management actions capable of promoting sustainable forest management and the preservation of critical ecosystem services. Prescribed fires as part of fire management could reduce fuel load and thus promote biodiversity conservation.

**Supplementary Materials:** The following supporting information can be downloaded at: https://www.mdpi.com/article/10.3390/fire6020077/s1. Figure S1: Land cover and land use maps of study area for 2001 (a), 2008 (b), and 2018 (c); Figure S2: Fire frequency (2001–2018) (a) and slope map category (b) for the study area; Figure S3: OLS Spatial autocorrelation report for dense miombo (a) and opened miombo (b); Figure S4: GWR Spatial autocorrelation report for dense miombo (a) and opened miombo (b); Table S1: Accuracy assessment of classification results based on confusion matrix for 2001, 2008 and 2018; Table S2: Accuracy assessment of the fire frequency map based on confusion matrix; Table S3: OLS and GWR results of the grid size cells tested under the parameters in analysis; Table S4: Summary of OLS Results–model variable_for dense miombo; Table S5: Summary of OLS Results-model variable_for opened miombo; Table S6: Geographically weighted regression (GWR) Results for dense miombo; Table S7: Geographically weighted regression (GWR) Results for dense miombo.

**Author Contributions:** Conceptualization, V.A.B., N.S.R., L.O. and R.R.B.; formal analysis, V.A.B., N.S.R., L.O. and R.R.B.; funding acquisition, N.S.R., L.O. and R.R.B.; methodology, V.A.B., N.S.R., L.O. and R.R.B.; supervision, N.S.R., L.O. and R.R.B.; validation, N.S.R., L.O. and R.R.B.; writing—original draft, V.A.B.; writing—review and editing, V.A.B., N.S.R., L.O. and R.R.B. All authors have read and agreed to the published version of the manuscript.

**Funding:** This study received financial support from Swedish International Development Cooperation Agency (SIDA), Grant number 51140073, Subprogram 123, and from Eduardo Mondlane University. Av. Julius Nyerere, 3453, Campus Universitário, Building #1, P.O. Box 257, Maputo, Mozambique.

**Informed Consent Statement:** Informed consent was obtained from all subjects involved in the study.

**Data Availability Statement:** The raw data supporting the conclusions of this article will be made available by the authors, without undue reservation.

**Acknowledgments:** We thank Célia Martins, Romano Guiamba and Ernesto Boane, for the field work support; Lund University Centre for Sustainability Studies, for the linguistic proofreading and financial support.

**Conflicts of Interest:** The authors declare that the research was conducted in the absence of any commercial or financial relationships that could be construed as a potential conflict of interest.

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
