# Peer review of "Exploring Spatial Distributions of Land Use and Land Cover Change in Fire-Affected Areas of Miombo Woodlands of the Beira Corridor, Central Mozambique"

_fire, doi:10.3390/fire6020077_

Round 1

Reviewer 1 Report

This paper has strong potential for publication in Fire.  At this point the manuscript needs revision.  The main changes are needed in how the results are presented, and how the results are discussed.  As written, the manuscript overemphasizes the statistical findings, without clearly or fully describing the real-world meaning of the results.  I have offered more specific comments below. 

  • p. 4: In the site description, please provide some basic floristic information on the miombo woodlands.  Which species are dominant?  Does species dominance vary over the study area? 
  • p. 4 and elsewhere: “Swidden agriculture” or “shifting agriculture” are preferable to “slash-and-burn”, due to negative connotations associated with “slash-and-burn”.
  • p. 5: Field observations were used to assess land cover types; what characteristics were observed?  What are the land cover types present in the area—the list given in the first paragraph of section 2.2.2?
  • p. 7: It would be helpful for the reader to state succinctly what all the reported statistics mean—not a full interpretation such as in the discussion section, but just a few words in plain English what the numbers mean.  Many readers who will be interested in the results may have lower understanding of the methods.  So, for example, “The Kappa index were [sic] 0.82, 0.81, and 86% [sic] for 2001, 2008 and 2018 respectively[, which shows strong agreement between datasets].”  Similar statements could be made for the various statistics, including in tables and figure captions, where relevant.
  • p. 7 and following: As seen in the quote above, there are several minor errors in the results section, and the writing generally seems less polished (e.g the second sentence in section 3.2 repeats content from the first sentence).  Review carefully to make sure that errors and unclear writing are found and corrected.
  • p. 14: In the first paragraph of section 4, provide a little more info on the historical context, specifically dates associated with important social changes, as well as some explanation of key social changes (why is it relevant that the refugees were in neighboring countries?).  The one date given, 1992, is almost a decade before the start of the period of analysis; did the 2001-2008 pattern emerge through the 1990s, or was it in place before then?  The really key date is 2008—what happened to produce such a clear change in pattern?  The authors cite an environmental education program, but it seems unlikely that this would have such clear, immediate impact on fire patterns.  Was there increased enforcement activities from forestry police that accompanied the environmental education activities?  The only other alternative given (lines 455-459) is that the forests/woodlands were widely eliminated.  What type of agriculture is practiced in such areas, based on my inference that there is no fallow land (i.e. woodlands) to claim in these areas?  
  • p. 14 and following: Put the results and discussion into plain English to the greatest extent possible.  Some of the sentences are jargon-heavy, and do not clearly communicate the results.  The ones on lines 471-476 are prime examples.  State the apparent real-world meaning of the findings, and refer to tables/figures for supporting statistics.  The lines 477-484 are clearer, to continue the example. 
  • p. 15: Regarding the “Slope has been described…” paragraph.  Fires spread more rapidly uphill than on flat areas, if other variables such as wind are constant.  There is a sizable literature on this.  Does this relationship affect the interpretation of the results?  Also, the fire literature in West African contexts show the importance of vegetation heterogeneity in affecting fire patterns within landscapes.  The West African results (such as reported by Paul Laris and collaborators) may not be directly applicable due to differences in geographic scale of analysis and data types, but what role might non-human variables other than slope have in observed fire patterns?  Overall, the discussion section needs to be reworked, to provided clearer explanation of the real-world meaning of and explanations for the results.

Reviewer 2 Report

The paper is interesting and covers an important topic however it currently needs work to make the language and structure clearer.  

General comments.

There seems to be some confusion at various points within the paper as to whether it is talking about land cover or land use change. They are sometimes but not always associated. Slash and burn land use, for example, can be conducted with minimal long-term change to land cover. It would be good to unlink land cover and land use change (LULCC) where appropriate and apply these terms more accurately in the context of their use.

The paper needs a good copy-edit to check the clarity of the writing and for typos.

concerned about the level of accuracy from global automated burnt area products when assessing fires from complex slash-and-burn agricultural landscapes. Although the paper suggests a reasonably high level of accuracy from the MCD64A1 product, many agricultural burns are small and may not be mapped by the 500m scale product. It would be interesting to have some reflection on further work that could be done using higher-resolution imagery to produce a fine-scale history of fire dynamics in the region. The MCD64A1 product may be predominantly capturing large wildfires rather than those associated with managed burning. Some discussion on how the paper could directly inform improved land management in the region would also be of interest.

Specific comments.

Lines 40-43

The rapid recovery of miombo woodlands after disturbance 40 allows them to act as a carbon sink, which is essential for climate regulation [9–11]. Areas 41 converted to agricultural land, following fallow, may be associated with a progressive 42 increase in carbon stocks [7,10].

Lines 49

Poverty in Sub-Saharan Africa is estimated to have declined slightly 49 from 53% to 51% between 1981 and 2005 [14].

This is refereing to landcover change not land use……

Line 53

Shifting cultivation and fire are among the key determinants of LULCC in the miombo 53 woodlands [15].

Line 63

Depending on the type of ecosystem and its adaptation, the correlation between fires and 63 LULCC can vary significantly, particularly in tropical regions, where fire frequency tends

Line 78-79. Slope effects spread speed but not nesseserily frequency.

Apart from influencing the 78 accessibility, slope is an important factor determining fire behaviour. Fires propagate easier 79 and faster upslope than on flat land, let alone downslope [29].

Line 89 – this appears to refer to 2016 data which is not current,

Currently, about 89 267,000 ha of forest is converted annually for shifting agriculture, settlement and infra-90 structure expansion, logging operations, and firewood and charcoal production [15,34].

Figure 1.

The legend colours don’t seem to match the map forest-type colours.

I am unsure what map (b) is showing, it does not seem to provide any further information. I

It would be good to be able to see the region in zoomed out context showing how it links Mozambique to Zimbabwe and other countries.

Other map figures would benefit from showing main roads as the main corridor linking the district and Zimbabwe is mentioned as a significant area of change.

Figure 5-6

The caption should be expanded, referring to each of the four sub-maps. Explain the legend colouring more clearly.

Lines 482-483 Citation typo

Lines 510-511

It seems from this section and subsequent discussion that access via the main Biera road corridor is the main driver of change. Maybe this could be explored as a variable in the analysis.  Also lines 542-543 suggest that road access is the key factor.

The last two paragraphs in the discussion seem more like background that could move to the introduction.

Round 2

Reviewer 2 Report

The manuscript is much improved. However, I would like to see figure 1 improved to give more space to the legend in relation to the map and border. It is currently cluttered.

It would also do one final read-through for language style. For example, the paragraph starting with the sentence here does not link well to the following sentence - I have provided an editing suggestion:

The spatially simultaneous occurrence of predictor variables involved in the LULCC indicates that fire occurrence in the region is influenced by humans, due to its strong link with agriculture and other activities, such as charcoal production. Uncontrolled use of fire as the main management tool, is a major contributor to fire occurrence in the miombo. These activities are a major source of uncontrolled fire in the miombo.
